



# Crop salinization by intense pumping in regional discharge areas of an inland aquifer system (Cenozoic Duero basin aquifer, Spain)

Esther Rodríguez-Jiménez[1, 2]; Pedro Huerta[3]; Laura Llera[4]; Pedro Carrasco[5], Clemente Recio[3]

[1] Facultad de Ciencias Forestales y del Medio Natural. Universidad Politécnica de Madrid, Spain.
5  [2] Confederación Hidrográfica del Duero. C/Muro, 5, 47004 Valladolid, Spain.
[3] Dpto. Geología, Facultad de Ciencias, Universidad de Salamanca, Pza. de la Merced S/N, 37008 Salamanca, Spain
[4] C/La Oliva, nº 11, Bajo E. 33300 Villaviciosa, Asturias, Spain
[5] Dpto. Ingeniería Cartográfica y del Terreno, Escuela Politécnica Superior de Ávila. Universidad de Salamanca, Av. Hornos Caleros, n° 50, 05003 Ávila, Spain

10  *Correspondence to*: Pedro Huerta (phuerta@usal.es)

**Abstract.** Salinization of crops irrigated with groundwaters in the Tordesillas area has been investigated to determine its cause. Hydrogeological, geophysical, and geochemical techniques reveal that regional saline groundwater flows through the Cenozoic aquifer system of the Duero Basin, discharging into the Tordesillas area. Groundwater salinity increases below 150-200 depth. TDEM profiles indicate that salinity distribution is influenced by mixing of local and regional flows, as well as by fault 15  structures affecting the Cenozoic succession. Isotopic analyses ($\delta^{18}O$, $\delta D$, $\delta^{34}S$) suggest multiple sources of dissolved sulphate and evidence that regional groundwaters recharged at higher altitudes and/or lower temperatures.

Irrigation return flows do not noticeably contribute to salinization, as $\delta^{18}O$ and $\delta D$ data from boreholes in the Duero Floodplain do not show any evaporation trend. Instead, intensive groundwater pumping during the irrigation season induces upwelling of saline groundwater. Piezometric records indicate that hydraulic potential at intermediate depths (about 100 m) decreases during 20  pumping at summer, facilitating upwelling of deeper saline groundwaters. Salinity profiles confirm this process, demonstrating a shift from fresher to more saline conditions over time.

Groundwater management authorities need to address this issue to prevent further salinization. These findings provide crucial insights for optimizing well design and identifying depths where groundwater is unsuitable for irrigation, ensuring sustainable water use in the region.

## 1 Introduction

Groundwater is an essential resource for farming and irrigated crops. Irrigation water must not contain contaminants that could affect human health and plants do not tolerate high salinity waters. During the summer of 2020 (and again in 2022) vegetable crops grown in the Duero floodplain nearby Tordesillas (Valladolid province, Northern Spain) died due to the high salinities of waters used for irrigation. Water was extracted from boreholes located in the Duero floodplain, screened at 100-150 m 30  depth. Similar vegetable crop die-offs had not been previously documented in the area, but it occurred again in 2022 season. According to Zaman et al. (2018) water electric conductivity (EC) values between 750–1500 µS/cm show detrimental effects



on vegetable crops, and values higher than 1500 µS/cm have adverse effects on many crops. Saline groundwater appears in aquifers by natural or anthropogenic causes. Common natural causes are connate saline groundwaters, marine intrusion, concentration by evaporation, geothermal origin, dissolution of evaporites or the incorporation of solutes from low-soluble

minerals for long periods (Chebotarev, 1955; Li et al., 2020). Anthropogenic causes are varied (Li et al., 2020) and those related to agriculture are the irrigation returns (Hibbs and Boghici, 1999) and intense pumping facilitating upwelling of saline groundwaters (Hibbs, 1998). Freshwater salinization is a growing problem in inland basins (Custodio, 2002; Thorslund et al., 2021). Intense withdrawal of groundwaters by pumping can exceed temporarily or on the long term safe salinity thresholds for irrigation in inland basins (Kaushal et al., 2005).

The aims of this research are to: 1) understand the origin of the saline groundwaters in Tordesillas area and document the natural behavior of the aquifer system in the area. 2) to evaluate if the anthropogenic (agricultural) activity is affecting crop salinization. To achieve these objectives different geological, hydrogeological, geophysical and geochemical techniques have been used. This article provides insight on the hydrogeology of the Cenozoic aquifer system of the Duero basin and should help farmers and the water management agency taking decisions to prevent the salinization problem in this basin.

**2 Hydrogeological setting**

The Cenozoic deposits of the Duero basin house a great aquifer system with an extension of 50000 km$^2$ (Fig. 1A). During the Cenozoic the basin was endorheic, characterized by alluvial deposits at its margins and lacustrine carbonate and/or evaporite deposits at the basin depocentres in several moments of its history. Paleogene deposits are preserved mainly at the basin margins, with large outcrops in the western zone (Salamanca-Zamora) and in the Almazán basin (Armenteros et al., 2002;

Ortega et al., 2022), which is connected with the Duero basin by Miocene rocks. Miocene strata, however, are the dominant material cropping out in the basin interior. Lithologically, these are mainly mudstones, sandstones and conglomerates in the basin periphery and limestones, marls and gypsum towards the basin centre. During the Pliocene the endorheic basin opened to the Atlantic Ocean and the early fluvial network started to incise and erode the Cenozoic succession (Antón et al., 2019; Cunha et al., 2019). These deposits conform a large aquifer system (up to 1200 m thick in the basin centre) characterized by

highly permeable detrital aquifers in its margins that pass basinward into a multilayer aquifer of more or less isolated sandstone and conglomerate beds within mudstones that behaves in most of the cases as aquitards. In the basin centre, along the present-day valleys of Duero, Pisuerga and Arlanzón rivers, different units of marls with abundant gypsum (more than 150 m thick) are present. These marls are aquitards which are covered by a karstified limestone aquifer (Calizas del Páramo) with an average thickness of about 20 m.

Regional groundwater flows from the recharge areas located mainly in the basin margins towards the discharge areas located mainly in the western – central part of the basin along the Duero river (CHD, 2009) (Fig. 1A). Deep brackish to saline groundwaters seep from springs or are intercepted by wells in mid-flow path areas where basement thresholds force the



groundwaters to rise, such as in the old spring of Las Salinas, in Medina del Campo (De la Hera-Portillo et al., 2021) or in the Villafáfila lakes (Armenteros et al., 2019; Huerta et al., 2021).

In the Tordesillas area (Fig. 1B) Miocene arkosic sandstones and mudstones crop out and towards the northeast these are covered by Miocene (Tortonian/Aragonian) marls with gypsum ("Facies Cuestas" unit) and by (Messinian/Vallesian) limestones ("Intermediate Paramo" unit). The Paleogene succession crops out 17 km SW from the town of Tordesillas. In this article we refer to the Paleogene and Miocene clastic deposits as the Cenozoic aquifer system which is a multilayer aquifer that in the area is >1000 m thick. Quaternary deposits consist of a sequence of staircase terraces that extend 18 km S from the

present-day Duero river. Thirteen conglomerate and sandstone terraces have been identified, from T1 located at +136/+128 m to T13 located at + 9/+3 m of the present-day Duero river (Rodríguez-Rodríguez et al., 2020). Upper Pleistocene to Holocene aeolian sand sheets occur on the lower terraces. The Duero river floodplain is made up of gravels, sandstones and siltstones (Fig. 1B). In this article the Quaternary deposits of the Duero river alluvial plain are referred to as the Quaternary aquifer, which is a non-consolidated detrital aquifer with a thickness that does not exceed the 20 m and lies unconformable onto the

Cenozoic aquifer system.

South of the Duero river (about 10 km apart), the relief shows a gentle slope formed by the staircase terraces with heights ranging from 750 m above sea level (m.a.s.l.) in the Rueda village area to 670 m.a.s.l. of the Duero river itself in Tordesillas. North of the Duero river the Miocene limestones (Intermediate Paramo Unit) form the top of a mesa (830 m.a.s.l.) that extends more than 140 km along the Duero-Pisuerga-Arlanzón river valleys.



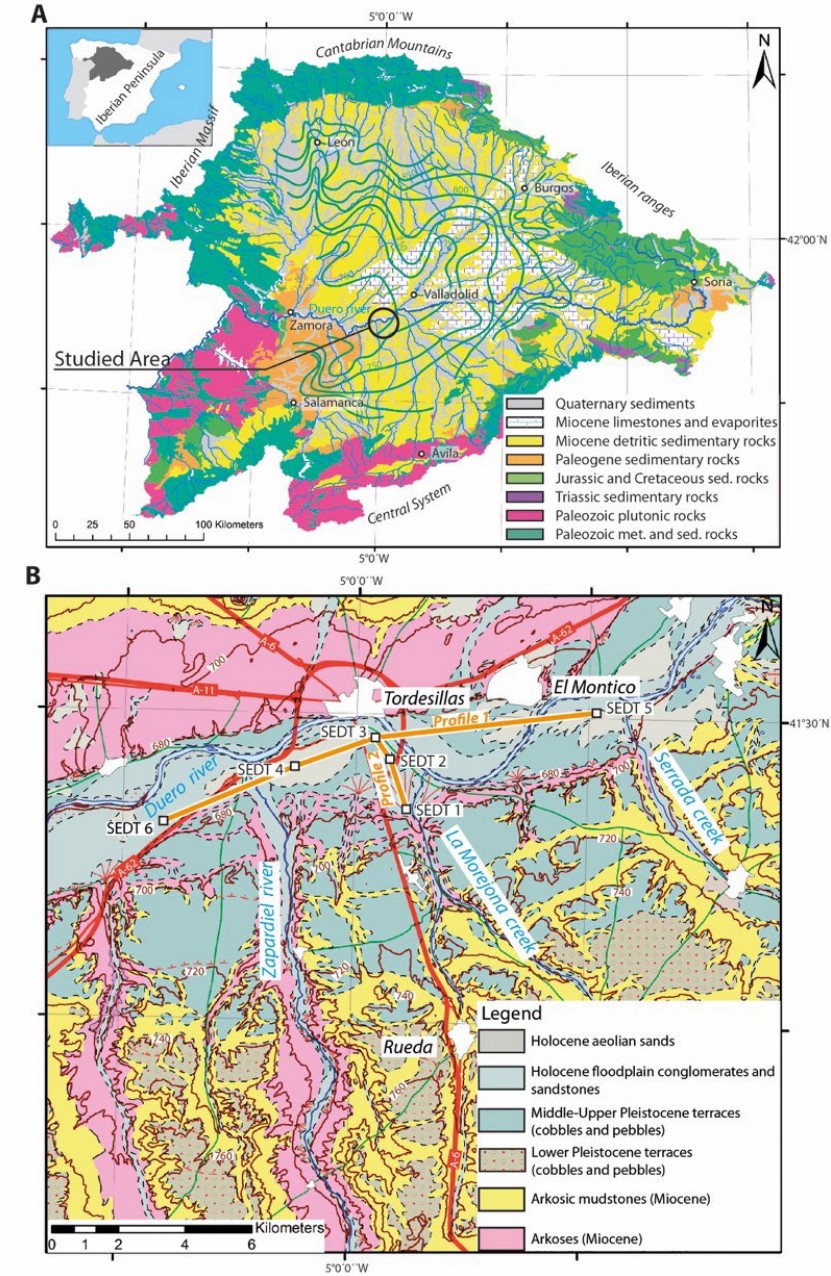

**Figure 1: Geological setting of the study area. A) Geology of the Duero river basin and regional piezometry (green lines) of the Cenozoic Duero aquifer system (reference piezometry, >200m) (CHD, 2009); modified from Huerta et al. (2021). B) Geological map of the Tordesillas area, modified from Pineda et al. (2008). SEDT profiles are located in the map.**




## 3 Climate and agriculture

Mean annual temperature in the Tordesillas area is 12.7 ˚C (data from Valladolid meteorological station, AEMET, 22 km away) (www.aemet.es). The lowest values occur during January with a mean temperature of 4.2 ˚C, while the highest are in July with a mean temperature of 22.3 ˚C. Average annual precipitation is 433 mm, normally concentrated from October to May. Summer precipitation is only occasional in the form of convective rains associated to short an intense storms, and evapotranspiration exceeds precipitation. Groundwater extraction for irrigation purposes intensifies during the summer. Most

of the agricultural activity in Tordesillas alluvial plain consist of vegetable crops like carrots, potatoes, vineyards, etc. Most groundwater extraction for irrigation comes from boreholes screened about 100-150 m depth. Overall groundwater extraction in the Tordesillas area amounts to about 0.65 Hm$^3$ per year, but most of the extraction concentrates along summer.

## 4 Methodology

Hydraulic heads were measured in dug wells (depths < 40 m) excavated in the Quaternary alluvial aquifer, and in several

boreholes and piezometers (depths > 40 m) of the control network of the hydrological administration (Confederación Hidrográfica del Duero, CHD) screened in the Cenozoic aquifer system. Hydraulic heads for the construction of piezometric maps were measured during the summer (July-August) of 2022 with a Solinst TLC 107 meter.

Groundwaters were sampled either by pumping or using a bailer. Some physico-chemical parameters like temperature, electric conductivity (EC) and pH were measured in situ with a portable multiparameter probe (Multi 340i of WTW®). Alkalinity was

measured in the field by titration and appropriate reagents. The sampling campaign was carried out during April 2022 in 12 sites (see Fig. 2). Samples were collected in screw-cap HDPE bottles filled to overflowing and stored refrigerated at 4 °C until analysis. Chemical analyses were done in the CHD laboratories. Major elements (Ca, Fe, Mg, Mn, K, Na) were analyzed with an ICP-MS (Agilent 7500 Cx). $SO_4^{2-}$, $NO_3^-$, $Cl^-$, $F^-$, $PO_4^{2-}$, were measured by ion chromatography with an Agilent HS/GC/MS chromatograph.

Hydraulic heads were monitored monthly in piezometers P-43 (screened between 34-70 m), P44 (98-190 m) and P42 (290 m) (Fig. 2) with a LevelSCOUT, Seametrics Ltd. from February 2021 to November 2022 (Fig. 3).

Vertical profiles of EC were measured in P42, P43 and P44 from May to September 2022 with a Solinst 107 TLC model and with a CT2X Seametrics Ltd. sensor. In piezometer P42, which is 290 m depth, a continuous record of temperature and electric conductivity was obtained using a TCME device from the company HIDROGEOMED. The sensor has a temperature precision

of ±0.4°C and 10 μS/cm EC.

Oxygen and hydrogen isotopic ratios were measured by Cavity Ring-down Spectroscopy (CRDS) using a Picarro L-2130i infrared laser spectrometer at the Stable Isotope Lab of the "Servicio Interdepartamental de Investigación" (SIdI), Universidad Autónoma de Madrid (www.uam.es/sidi). Normalization of data used internal references (usually waters remaining after International Atomic Energy Agency IAEA-led round robin tests) interspersed among unknowns. Additionally, each batch

included two or more vials with V-SMOW (Vienna Standard Mean Ocean Water) and one with SLAP (Standard Light



Antarctic Precipitation), and usually another one with GISP (Greenland Icesheet Precipitation). Long-term reproducibility ($1\sigma$) is better than ±0.03‰ for oxygen and ±0.15‰ for hydrogen isotopic ratios. Isotopic data of local rain from Valladolid were obtained from the "Red de Vigilancia de Isótopos en Precipitación (REVIP), (CEDEX)" that is part of the Global Network of Isotopes in Precipitation (GNIP) of the IAEA (the data corresponds to the period 2000-2016).

Water samples for $\delta^{34}S_{SO4=}$, $\delta^{18}O_{SO4=}$ analyses were filtered through 0.45 µm nylon filters. Water-soluble sulphate was precipitated by addition of 10% $BaCl_2$ solution to the previously acidified (to pH≈2) water, essentially following the original method of Rafter (1967), as amended by Sakai and Krouse (1971) and Mizutani and Oana (1973). The $BaSO_4$ precipitate was separated from the solution by filtering through Whatman #42 filter (125 mm diameter), washed with distilled water, and air-dried. Sulphur isotope analyses ($\delta^{34}S$) were done on $SO_2$ produced off-line following the method of Robinson and Kusakabe

(1975), with modifications for sulphates by Coleman and Moore (1978). Isotopic ratios were determined on a dedicated dual inlet SIRA-II (VG-Isotech) mass spectrometer. The oxygen isotopic composition of the $BaSO_4$ ($\delta^{18}O_{SO4=}$) was measured on CO obtained by pyrolysis on a EuroVector EA3000 elemental analyzer, coupled on-line to an Isoprime continuous flow mass spectrometer. Isotopic ratios are reported in the usual "$\delta$" notation as per mil (‰) deviations from V-SMOW ($\delta^{18}O_{H2O}$, $\delta D_{H2O}$, $\delta^{18}O_{SO4=}$) or CDT ($\delta^{34}S_{SO4=}$).

The TDEM survey was carried out during May 2022. Soundings were placed aligned in a ENE-SSW trend, along the valley of the Duero river (SEDT-3 to 6) to construct geoelectric profile 1 (Fig. 4A). Soundings SEDT-1 to 3 are aligned in a NNW-SSE trend, from the terraces at the south of the Duero rive to the present day floodplain, to construct geoelectric profile 2 (Fig. 4B). The surveys were carried out by the company Técnicas Geofísicas S.L. with a TerraTEM (MONEX GeoScope Ltd.) transient electromagnetic survey system. The array consisted on 200 x 200 m square loops in the mode of coincident loops.

Several 10 ms measurements were performed in every site, varying the number of stacks (up to 3000 stacks) to minimize electromagnetic noise. The methodology used allowed reaching a depth of exploration of about 400-500 m. Processing of the data obtained and its conversion from apparent to real resistivities were made with IX1D V3 (INTERPEX ltd.). A smooth model was generated for every TDEM site measured. Occam´s inversion (Constable et al., 1987) was used to obtain the 30-40 layers smooth model.

**5 Results**

**5.1 Piezometry**

**5.1.1 Quaternary aquifer**

The Duero-river alluvial plain aquifer in the Tordesillas area is connected with the Duero river. TFrom San Miguel del Pino to the East to Torrecilla de la Abadesa to the west, the river is a gaining river as evidenced by the piezometry map of the

Quaternary aquifer (Fig. 2). Piezometric gradient along the river is 0.001 and perpendicularly to the alluvial plain it is about 0.006. Hydraulic head at the east, upstream of the mentioned area, is 675 m.a.s.l. while it is 662 m.a.s.l downstream towards



the west. Groundwater table depths vary from 1.86 m in P-35 to 11.33 m in P-70, and the mean depth is 3.43 m. Seasonal changes in water table depth from spring 2021 to summer 2022 are minor, with mean difference between maximum and minimum of 0.59 m.

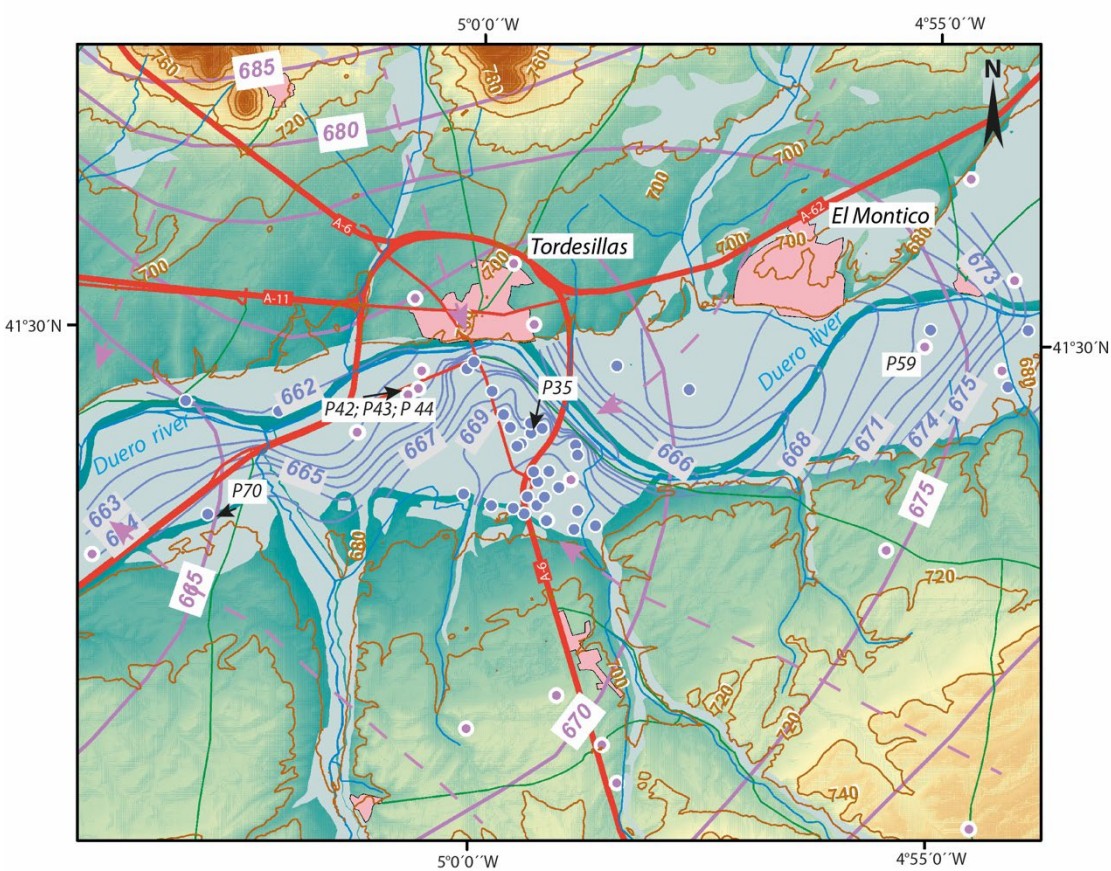


**Figure 2: Water table map on a digital elevation model of the Tordesillas area for the Quaternary aquifer (blue thin lines) and the Cenozoic aquifer system of the Duero basin (purple thick lines). Arrowed-dashed purple lines are the regional flow lines. Blue dots are dug wells (depths < 40 m) and purple dots are boreholes (depths > 40 m). Wells mentioned in the text are labeled in the map. DEM has been provided by the Instituto Geográfico Nacional (www.ign.es).**

**5.1.2 Cenozoic aquifer**

The Cenozoic Aquifer of the Duero basin in Tordesillas discharges into the Duero river favoring the evacuation of the shallow and deep groundwaters (Fig. 1A). Regional and local piezometry measured in boreholes with depths greater than 40 m show a flow component from east to west and towards the Duero river (Fig. 2). Hydraulic heads are highest both north and south of the river, and lowest to the west alongside the river valley. In the boreholes (screened at depths higher than 40 m) located

within the river valley hydraulic head is higher than that measured in dug wells (screened at depths shallower than 40 m) of the Quaternary aquifer (Fig. 2). In P42, P43 and P44 piezometer set, screened at depths of 290 m; 34-70 and 98-190 m respectively, the historic record (1996-2022) (Fig. 3A) shows seasonal oscillations in hydraulic head. Before 2012 hydraulic




head in P42 (290 m) was higher than in P43 (screened between 34-70 m). Focusing on two-year data (January 2021 - December 2022) (Fig. 3B) a similar seasonal trend for P43 and P44 can be observed. P43 and P44 (34-70 m and 98-190 m depth

respectively) have higher hydraulic heads during winter (December to March) and lower ones in summer (June to September). Hydraulic head is always higher in P43 than in P44, but the difference increases during summer. Seasonal hydraulic head oscillation is about 8 m in P43 and about 13 m in P44. The record of P42 is out of phase with respect to that of P43 and P44. The maximum hydraulic heads are recorded in June and the minimum during October. From April to October the hydraulic head in P42 is higher than in P43 and P44. On the contrary, from October to March hydraulic potential in P42 is lower than in

P43 and P44. Other deep borehole (P59) located on the alluvial plain upstream of the studied area is a fluent borehole during the most of year but during summer hydraulic head decreases below terrain surface.

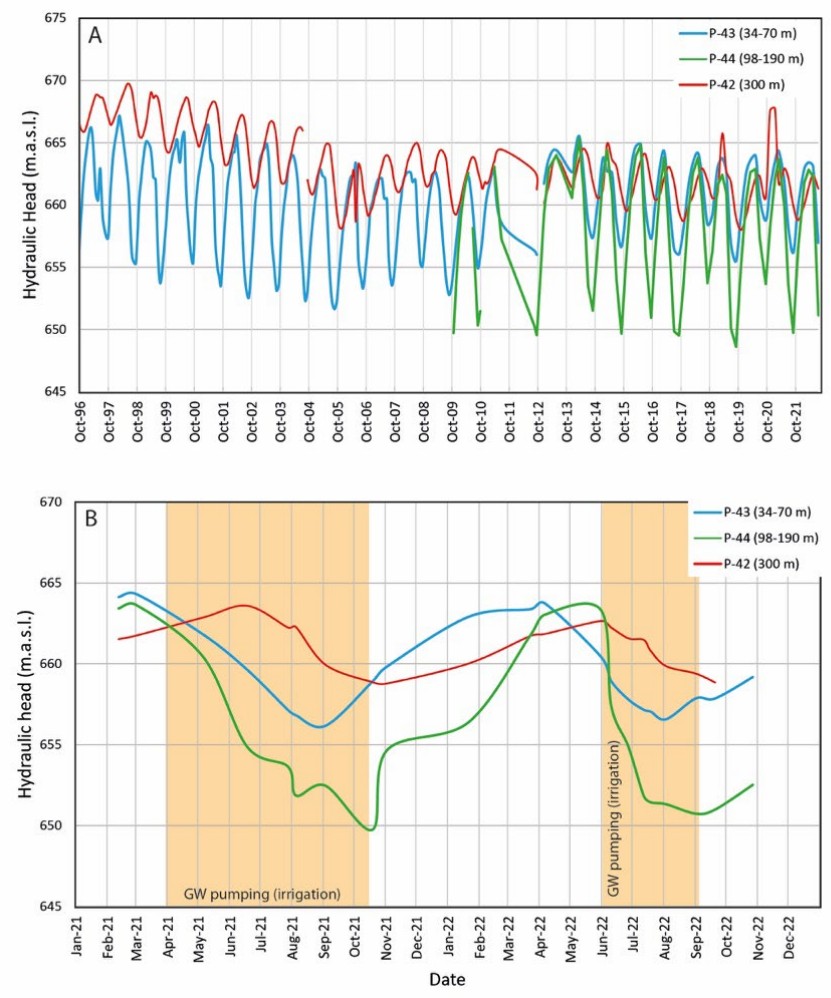

**Figure 3: Hydraulic head variations in piezometers P42, P43 and P44 (see location in Fig. 2). A) Seasonal hydraulic head variations for the period October 1996-October 2021. B) Detailed hydraulic heads for the period January 2021 to December 2022 with**
**indication of the irrigation period.**





## 5.2 Groundwater salinity prospection

### 5.2.1 Geophysics results

Three TDEM soundings (SEDT 1, 2, 3) were placed perpendicular to the Duero river valley in its left bank (south) (conforming profile 2) and four (SEDT 5, 3, 4, 6) are aligned parallel to the river from east to west  profile 1) (Figs. 1B and 4). The depth explored in these soundings is about 550 m from the ground surface, reaching a height of 100 meters above sea level (m.a.s.l.). Four geoelectric units (GU) have been defined as a result. GU-1 is a high resistivity unit (from 150-100 Ohm.m to about 20 Ohm.m) that appears at the top of the profiles. Its thickness varies from 90-80 m to 5 m. GU-2 appears below GU-1 and shows resistivities from 20 to 8 Ohm.m. Its thickness varies from 260 m in SEDT-3 to 20 m in SEDT-5. GU-3 locates below GU-2 and displays the lowest resistivity values from 8 to 4 Ohm.m. In profile 2 and in SEDTs 3 and 4 of profile 1 it appears at the base of the explored depth. GU-4 appears below GU-3 in SEDTs 5 and 6. It is a high resistivity unit with values ranging from 25 to 10 Ohm.m. In SEDT 6 appears below 420 m.a.s.l. to the base of the profile where it reaches the highest resistivity values. In SEDTs 3 and 4 is probably below the explored depth (Fig. 4).

Profile 1 (Fig. 4A) is 14 km long, and parallel to the Duero river valley. All the SEDT stations are located on the alluvial deposits of the Duero river. GU-1 is thicker at the north east in SEDT 5, thinning progressively towards the southwest. GU-2 reaches a large thickness in SEDTs 3 and 4 and reduces drastically towards the southwestern and northeastern ends of the profile (SEDTs 5 and 6). In those areas where GU-2 has a reduced thickness, the low-resistivity GU-3 is closer to the ground surface, and a higher resistivity unit (GU-4) appears below. In contrast, in the central parts of the profile (SEDTs 3 and 4) GU-3 is thicker and there is no evidence of GU-4 along the 550 m of depth explored.

Profile 2 (Fig. 4B) is 2.4 km long and perpendicular to the river valley on its left bank. The southern end of the profile (SEDT-1) is located on more elevated terrain than the others. In this southern end GU-1 is thicker and wedges laterally northwards. The thickness of GU-2 is about 200 m and slightly less in SEDT-1. GU-3 appears at the bottom of the whole profile below 400 m.a.s.l.



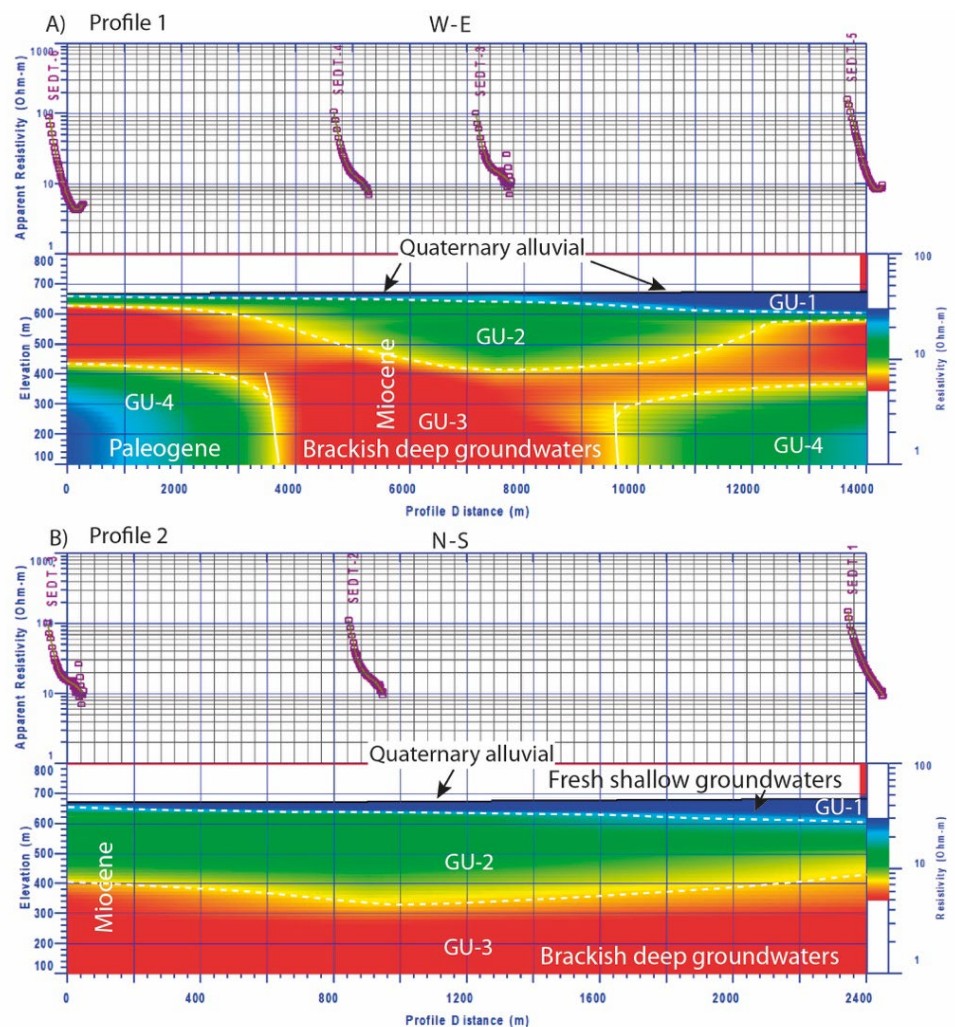

**Figure 4: Time Domain Electromagnetic profiles (TDEM) in Tordesillas area. See location of the SEDTs in Fig. 1B. A) Profile 1; B)**
**Profile 2.**

### 5.2.2 Electric conductivity and temperature results

EC vertical profiles were measured in P42, P43, and P44 (Fig. 5). These piezometers are close to SEDT-4. EC vertical
variations in P43 (screened from 34-70 m depth) show an increase from 500 to 1500 µS/cm at depths of 10 m in spring (from
April to June) and at depths about 25 m in August and September. P44 (screened from 98-190 m depth) records constant EC
values of 950 µS/cm for the first 90 m (570 m.a.s.l.) increases rapidly from then on to 3500-4000 µS/cm. In the record of
August 2022 there is a sharp EC increase at 140 m (520 m.a.s.l.) reaching 5500 µS/cm. In September the EC shift locates
shallower, at 126 m depth (554 m.a.s.l.). In piezometer P42 (screened at 290 m depth), at the end of June 2022, EC was about
3000 µS/cm till 140 m depth (520 m.a.s.l.). From 140 to 160 m depth (520-500 m.a.s.l.), EC increases from 3000 to 7000
µS/cm. From 280 to 290 m depth (380-370 m.a.s.l.) EC increases progressively reaching 13000 µS/cm (Fig. 5). The increase





in groundwater EC observed in P42 from about 2800 µS/cm to about 6000 µS/cm occurs at depths of 140-160 m (520-500 m.a.s.l) in June, but at shallower depths of 100-110 m (560-550 m.a.s.l.) in July and even less (about 85 m depth; 575 m.a.s.l.) in September.

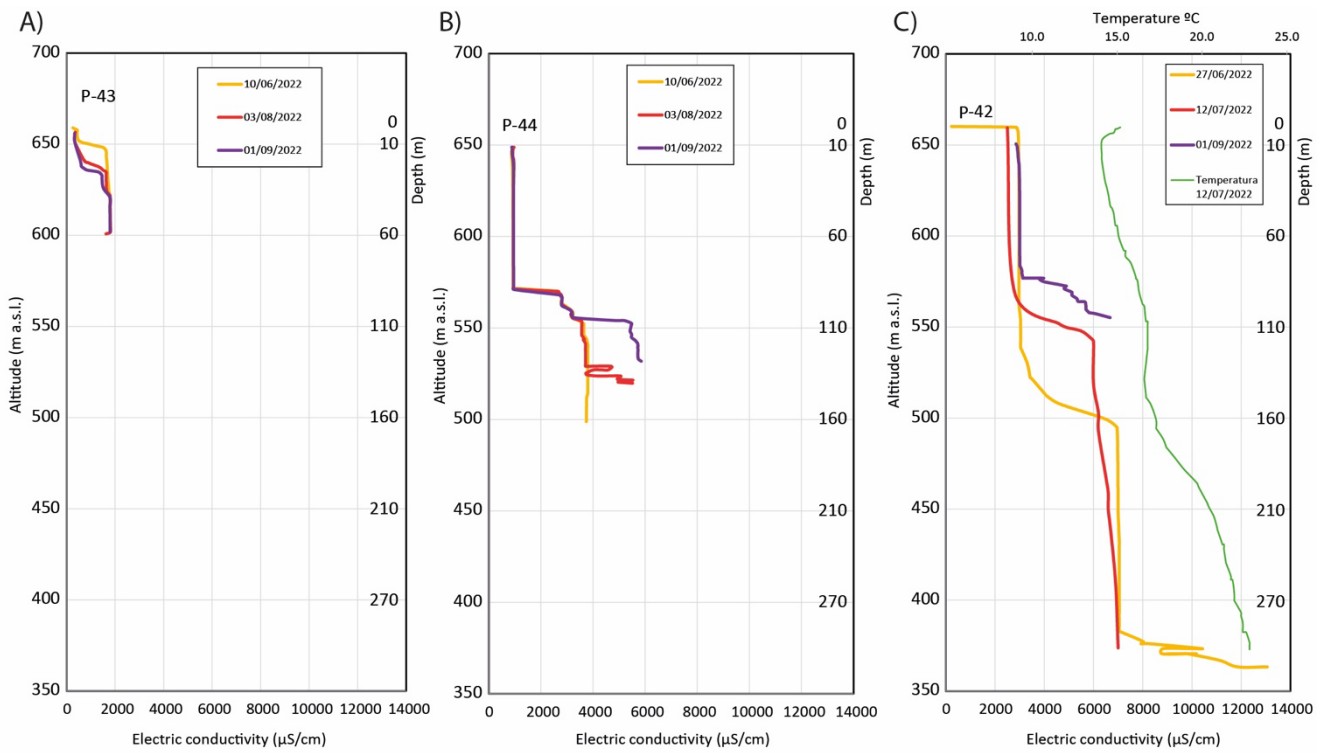

**Figure 5: Vertical Electric conductivity profiles measured along summer 2022 in A) P43 (screened from 34-70 m depth); B) P44**
**(screened from 98-190 m depth); and C) P42 (screened at 290 m depth). This piezometer includes temperature record.**

Temperature in P42, at 8.2 m depth was 18.2 °C but reduced rapidly to 14.2 °C at 18 m depth (Fig. 5). From 18 m to 120 m shows a progressive increase of temperature reaching 16.8 °C (temperature gradient 0.025). From 120 to 150 m depth temperature decreases less than 1°C. Below 150 m depth temperature increases progressively reaching 22.5 °C at 290 m depth. The mean temperature gradient is 0.03.


### 5.3 Groundwater chemistry

Different samples from dug wells and boreholes located in the Duero river floodplain, boreholes outside the floodplain and one sample from a surface pond were analyzed to know their major element chemical composition (Table C1; Fig. 6). Groundwater samples from boreholes located in the Duero floodplain are dominated by the Na- $HCO_3^-$-Cl hydrofacies (Fig.
6). Groundwaters from boreholes located outside the floodplain are mainly Ca-$HCO_3^-$ in composition, although some samples have intermediate anion proportions between bicarbonate and chloride (Fig. 6). The salinity of the groundwaters sampled in




boreholes located within the floodplain (300-2800 mg/L) is generally higher than those sampled in boreholes outside the floodplain (160-720 mg/L) (Fig. 7A). Groundwaters sampled from dug wells which are screened in the Quaternary alluvial aquifer varies from Na to Ca in cation composition and Cl or not dominant anion content. (Fig. 6). Samples from dug wells show mainly high salinities and one sample reaches 5000 mg/L (Fig. 7A). Nitrate content is also high in these dug well samples reaching 210 mg/L (Fig. 7A). The Na+K/Na+K+Ca vs Cl/Cl+HCO$_3^-$ ratio (Fig. 7B) shows that: A) groundwaters dominated by calcium and bicarbonate, occur mainly at the boreholes outside the floodplain; B) Na-HCO$_3^-$-Cl groundwaters are sampled mainly in boreholes within the floodplain and also outside the floodplain, and C) intermediate compositions appear in dug wells excavated on the floodplain.

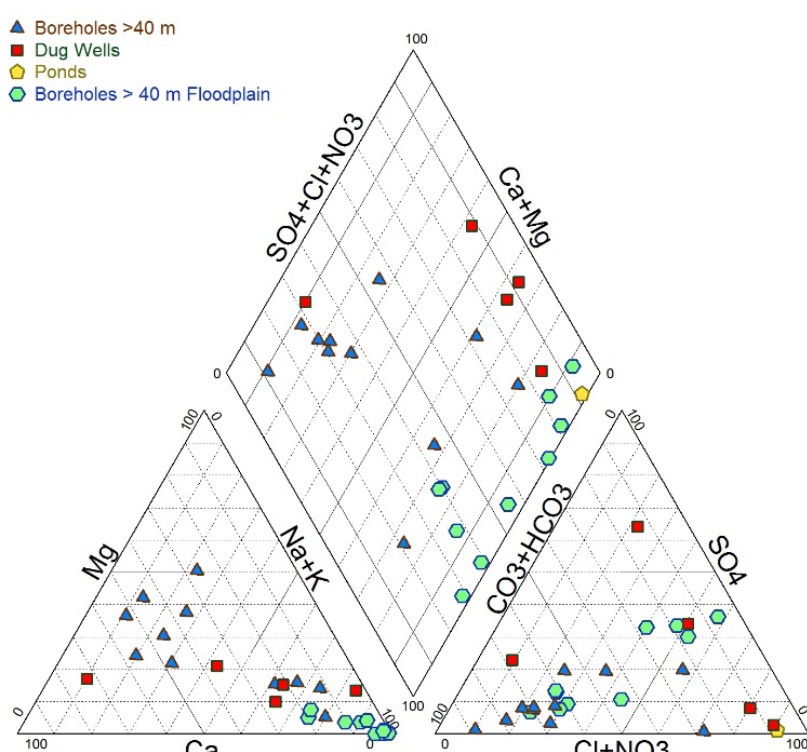

**Figure 6: Piper diagram for the groundwater samples analysed in Tordesillas. See data in Table C1.**



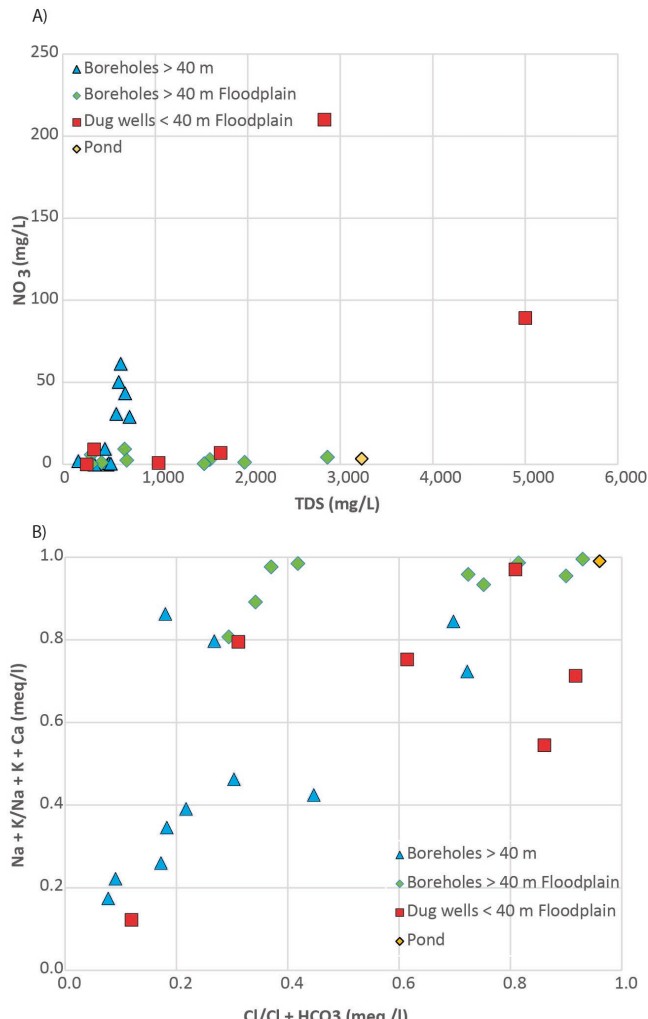

**Figure 7: Relationships between A) [NO$_3^{2-}$] and TDS; B) [Na$^+$]+[K$^+$]/[Na$^+$]+[K$^+$]+[Ca$^{2+}$] and [Cl$^-$]/[Cl$^-$]+[HCO$_3^-$].**

## 240   5.4 Isotopic geochemistry

Groundwaters analyzed were sampled from dug wells screened at depths lower than 15 m and in boreholes or piezometers screened at depths larger than 40 m. Three data of surface waters were sampled in the Duero river and in a pond within the Duero floodplain. Isotopic composition of local precipitation is publicly available data from REVIP (CEDEX) / GNIP (IAEA) for the Valladolid meteorological station (AEMET) (monthly means for the period 2000-2016): $\delta^{18}O_{VSMOW}$ varies from -14.03
to -2.82 ‰ and $\delta D_{VSMOW}$ from -103.9 and -15.5 ‰ obtained in February 2006 and August 2000 respectively. A Local Meteoric Water Line (LMWL) is defined as $\delta D = 7.47 \, \delta^{18}O + 3.7$, which lies very close to the Global Meteoric Water Line (GMWL), as defined by Craig, (1961), but a little bit below it (Fig. 8A).Amount-weighted average mean precipitation is $\delta^{18}O_{VSMOW}$ -7.6 ‰ and $\delta D_{VSMOW}$ -52.31 ‰ (n= 163) for the period 2000-2016.





δ¹⁸O$_{VSMOW}$ and δD$_{VSMOW}$ of two samples from the Duero river (M1 and M2) (Table C2), plot very close to the average mean

precipitation in Valladolid. The sample from the pond shows the heaviest values measured (δ¹⁸O$_{VSMOW}$= -1.12 ‰ and δD$_{VSMOW}$

= -9.7 ‰), as expected, and separates from the LMWL (Fig. 8A).

Isotopic values of the samples obtained from boreholes range from -9.74 to -8.09 ‰ δ¹⁸O$_{VSMOW}$ and from -71.7 to -60.5 ‰

δD$_{VSMOW}$. They are lighter than the both average mean precipitations in Valladolid and the Duero river samples. Borehole

groundwater isotopic values are similar to those from borehole groundwaters reported North of the Duero river, in the

Villafáfila area (Huerta et al., 2021). Isotopic values from dug wells, although there are only two samples, lie in the heavier

part of the range of those of the boreholes (Fig. 8A).

Even when dug wells are excavated in the Quaternary alluvial aquifer that is connected to the Duero river, isotopic values are

more similar to groundwaters from boreholes than to those from the river.

Isotopes from borehole groundwater-SO$_4^=$ have δ¹⁸O$_{VSMOW}$ values ranging from 10.25 to 17.6 ‰ and δ³⁴S$_{CDT}$ from 10.4 to

17.06 ‰ (Table C2). Two samples from the Duero river have δ¹⁸O$_{VSMOW}$ = 11.24 and 12.8 ‰ and very similar δ³⁴S$_{CDT}$ = 12.3

and 12.4‰ respectively. One sample from dug well P28 has δ¹⁸O$_{VSMOW}$ = 12.12 ‰ and δ³⁴S$_{CDT}$ = 8.8 ‰, which is the lowest

value of any sampled water (Fig. 8B).

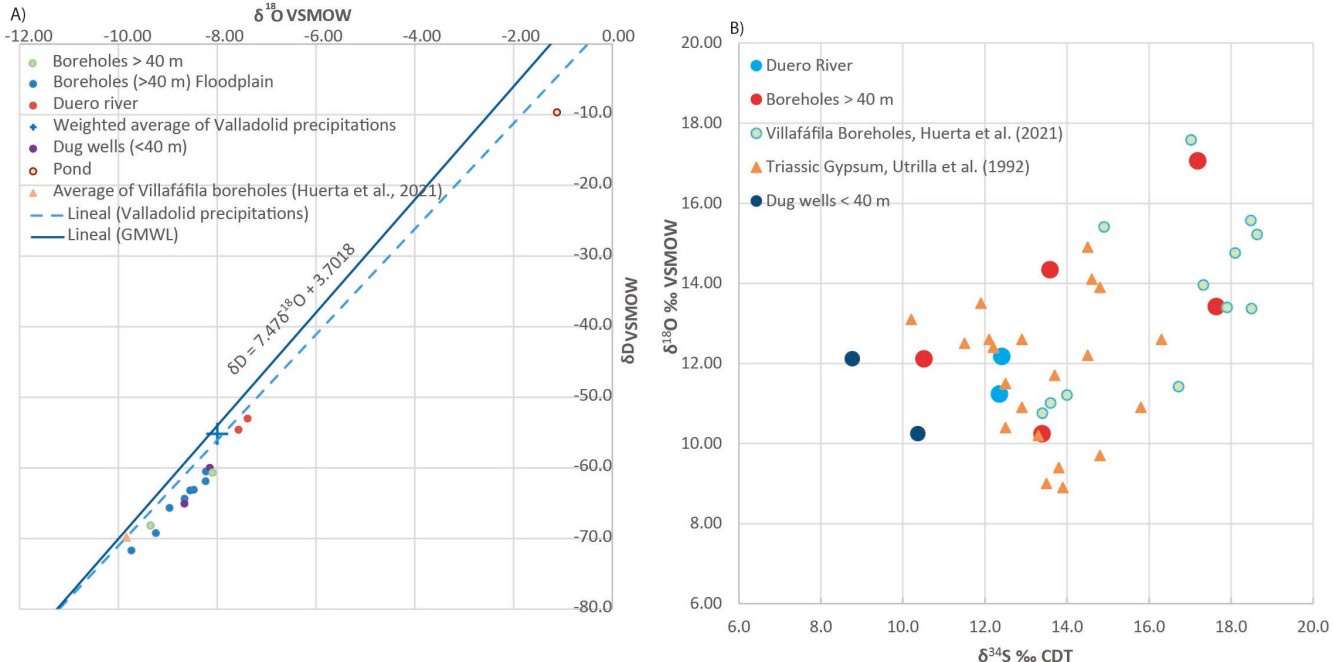

**Figure 8: Stable isotope composition A) δD$_{VSMOW}$ Vs δ¹⁸O$_{VSMOW}$ of groundwaters and the Duero river. GMWL and the local**
**Valladolid precipitation are included for reference, as is average Villafáfila data (Huerta et al., 2021). B) Dissolved-sulphate isotopic**
**composition δ¹⁸O$_{VSMOW}$ Vs δ³⁴S$_{CDT}$ of groundwaters and the Duero river. Data from Villafáfila Boreholes (Huerta et al., 2021) and**
**from Triassic gypsums (Utrilla et al., 1992) are included for reference.**



## 6 Discussion

Irrigated crops in the Tordesillas area were lost in 2020 and 2022 because of irrigation with saline groundwaters drawn from
boreholes. Below we discuss how different data gathered shed light on the problem, and help us understanding the causes of
salinization.

### 6.1 Salinity distribution and aquifer structure

Deep groundwaters in the Tordesillas area are of the Na- $HCO_3^-$-Cl type, with  EC ≈7100 µS/cm at depths of 100-250 m,
although can reach up to 13000 µS/cm at higher depths, as observed in piezometer P42 (Fig. 5). This latter, extremely high
value should be taken with caution: it was measured just once, in June 2022, when we were trying to reach as deep as possible
with the probe, and it is uncertain if mud resuspended from the hole bottom might have deleteriously influenced the measure
taken. TDEM profiles reveal that low resistivities dominate in the deepest parts (GU-3) which is interpreted as saline
groundwaters. This is evident along profile 2 (N-S) and in SEDTs 3 and 4 in profile 1 (W-E) (Fig. 4). Laterally in SEDTs 5
and 6 higher resistivity GU-4 appears below GU-3. The different geoelectric units are interpreted as follows: GU-1 is
interpreted as the Quaternary alluvial aquifer installed onto the Duero floodplain deposits, the aeolian sand sheets, and the
stair-case terraces, as well as the upper part of the Miocene succession. Groundwaters in GU-1 are freshwaters. This is similar
to what has been observed in the Villafáfila area, north of the Duero river (Huerta et al., 2022). Thickness reduction in GU-1
along profile 2 from SEDT 1 to SEDT 3 (S to N) can be interpreted as a combination of higher rainfall recharge on the terraces
(South) and increased groundwater extraction, and mixing with more saline groundwaters, in the floodplain areas near SEDT
3 (North) (Fig. 4B). This is consistent with the salinity of groundwaters measured in dug wells located in the floodplain. The
thicker part of GU-1 in profile 2 (SEDT 5) is interpreted as the shallow groundwater input from the Serrada creek. GU-2 is
interpreted as the transition zone between the shallow, fresh groundwaters of GU-1 and the saline regional flows of GU-3.
GU-2 corresponds with the upper-middle zones of the Cenozoic aquifer (Miocene). GU-3 is interpreted as the regional saline
groundwaters that occupy the deepest parts of the Miocene aquifer in the discharge areas of the Duero basin (Fig. 4). GU-4 is
below GU-3 in profile 1 and has higher resistivities. GU-4 is interpreted as the Paleogene units of the Cenozoic succession
which crop out towards the west of the Tordesillas area (Fig. 4A). GU-4 groundwater salinities are lower than those in GU-3
(Miocene succession) probably due to the contribution of recharge on the Paleogene units and their reduced flow path lengths
and residence times. Abrupt variations in the depth of the base of GU-3 observed in profile 1 between SEDTs 6 and 4, and 3
and 5, are interpreted as extensional faults controlling the thickness of the Miocene succession, which increases towards the
basin center (NE of Tordesillas) (Fig. 4A). The possible fault existing between SEDTs 6 and 4, could be coincident with the
valley of the Zapardiel river, which is aligned in NNW-SSE direction.

Regional piezometric maps for the Cenozoic aquifer system of the Duero basin (Figs. 1A and 2) indicate that recharge areas
locate in the basin margins and the discharge areas for regional groundwater flow locate mainly along the Duero river valley
(IGME-CHD, 2008; IGME, 1980b; Porras Martín, 1973). From recharge towards discharge areas there is an increase in



groundwater salinity (IGME, 1980a) that can be observed, from south to north in wells of the southern part of the aquifer system (Nieto et al., 2020). The vertical upwards decrease in salinity in the Tordesillas area (increase in resistivity in the SEDT profiles, and decrease in EC in the well logs) is consistent with the discharge of deep regional groundwaters in this area (Figs. 2 and 4). Saline regional groundwaters progressively mix upwards with less saline groundwaters of shorter or local trajectories and with the fresh alluvial groundwaters. This produces a zone with intermediate salinities located at intermediate depths (80-200 m). This zone is characterized by GU-2 in the TDEM profiles and the wells used for irrigation are screened in this zone (Fig. 4).

The comparison of the piezometry of the Quaternary aquifer and that of the Cenozoic aquifer system points to an upwards component of the groundwater flow due to the higher hydraulic potential of the latter (Figs. 2 and 3A). This is confirmed by the occurrence of flowing wells located in the floodplain of the Duero river and the piezometric record of the piezometers P43, P44, and P 42 which are screened increasingly deeper (Fig. 3A).

**6.2 Role of irrigation returns in salinization**

One of the causes of groundwater salinization in non-coastal aquifers is related to continuous solute recycling due to irrigation returns (Foster, 2022; Yakirevich et al., 2013). Shallow groundwaters sampled from dug wells excavated in the Quaternary alluvial aquifer have TDS ranging 260-5000 mg/L (Table C1). Nitrate concentration in these groundwaters is between 85 and 210 mg/L, while most deep groundwaters (Boreholes > 40 m floodplain) have less than 9 mg/L (Table C1; Fig. 7A). An increase in nitrate concentration in shallow groundwaters is a good proxy to identify salinization by irrigation returns (Pulido-Bosch et al., 2000). Widespread nitrate pollution of shallow groundwaters may have diverse origins, such as the use of chemical fertilizers, atmospheric deposition, leaking sewers, seepage of slurry from landfills, and many others (Pastén-Zapata et al., 2014; Wakida and Lerner, 2005). Agricultural fertilizers, however, are most relevant in the area being considered. Water stable isotopes should be good indicators of salinization by solute recycling produced by irrigation returns since as groundwater evaporates and concentrates salts, its δD and δ¹⁸O become heavier (Rizvi et al., 2023; Wang et al., 2016), aligning along an evaporation line with lower slope than the local meteoric line on a δD vs. δ¹⁸O plot. This feature has been identified in different aquifers during the dry season (Duque et al., 2011) as well as in arid areas (Liu et al., 2018). In the Tordesillas area, however, the δ¹⁸O and δD from boreholes and dug wells do not fit such an evaporation line (Fig. 8A). Measured values are lighter than the weighted average of precipitation in nearby Valladolid Meteorological Station or those of the Duero river. We interpret that the high salinities measured in dug wells located in the Duero floodplain (up to 5000 mg/L TDS) are produced by the solute contribution from irrigation wells (boreholes > 40 m) to the Quaternary aquifer groundwaters (Table C1; Fig. 7A). This is consistent with the δ¹⁸O and δD signal from dug wells, which lies within the range of borehole values (Fig. 8A), and is also evidenced by the intermediate Na-Ca contents measured in dug wells. This supports the idea of a mixed contribution of Na-HCO₃⁻-Cl groundwaters from boreholes located in the floodplain, and Ca-HCO₃⁻ groundwaters measured in boreholes outside the floodplain which intercepts local groundwater flows (Figs. 6 and 7B). The absence of an evaporation trend in δ¹⁸O and δD





from boreholes and dug wells evidences that solute recycling by irrigation returns is not the origin of the salinization problems in the Tordesillas area.

### 6.3 The role of pumping in crop salinization

Seasonal changes of hydraulic head observed in piezometers P43 (screened between 34-70 m), P44 (98-190 m) and P42 (290 m) indicate that there is an important variation along the year (Fig. 3A). The greatest amplitude was observed in P44, which varies 13 m from winter to summer (Fig. 3B). This occurs because pumping wells for irrigation are extracting groundwater from depths of 100-150 m. Piezometric variation observed in P43 shows a similar trend to P44 but with shorter oscillations (about 7 m for the period 2021-2022) (Fig. 3B). This similar response of the hydraulic potential in the upper part of the

Cenozoic aquifer is interpreted as a rapid groundwater transfer towards the pumped zones because of high effective porosity and hydraulic conductivity. Piezometric oscillations of P42, however, are out of phase with those of P43 and P44, which is interpreted as a slower response of the lower part of the aquifer to pumping of middle parts. During the irrigation campaign, hydraulic head in the deeper parts of the aquifer is higher than in shallower parts (Fig. 3). This intensifies the upwards groundwater flow in the Tordesillas area. Vertical EC profiles measured in P43, P44, and P42 throughout the summer indicate

that the shift observed in EC occurs at shallower depths from June to September (Fig. 5). This salinity rise is interpreted as being produced by intense groundwater pumping for irrigation throughout the summer (Fig. 9). The increase in EC reaches 6000 μS/cm at the depths where pumping wells are screened (100-150 m). Pumping of these saline regional groundwaters produces the loss of vegetable crops. Intense pumping has produced the upwelling of saline groundwaters and the lowering of groundwater table in aquifers like the Hueco Bolson in El Paso/Ciudad Juarez corridor (US/Mexico border) (Hibbs, 1998).



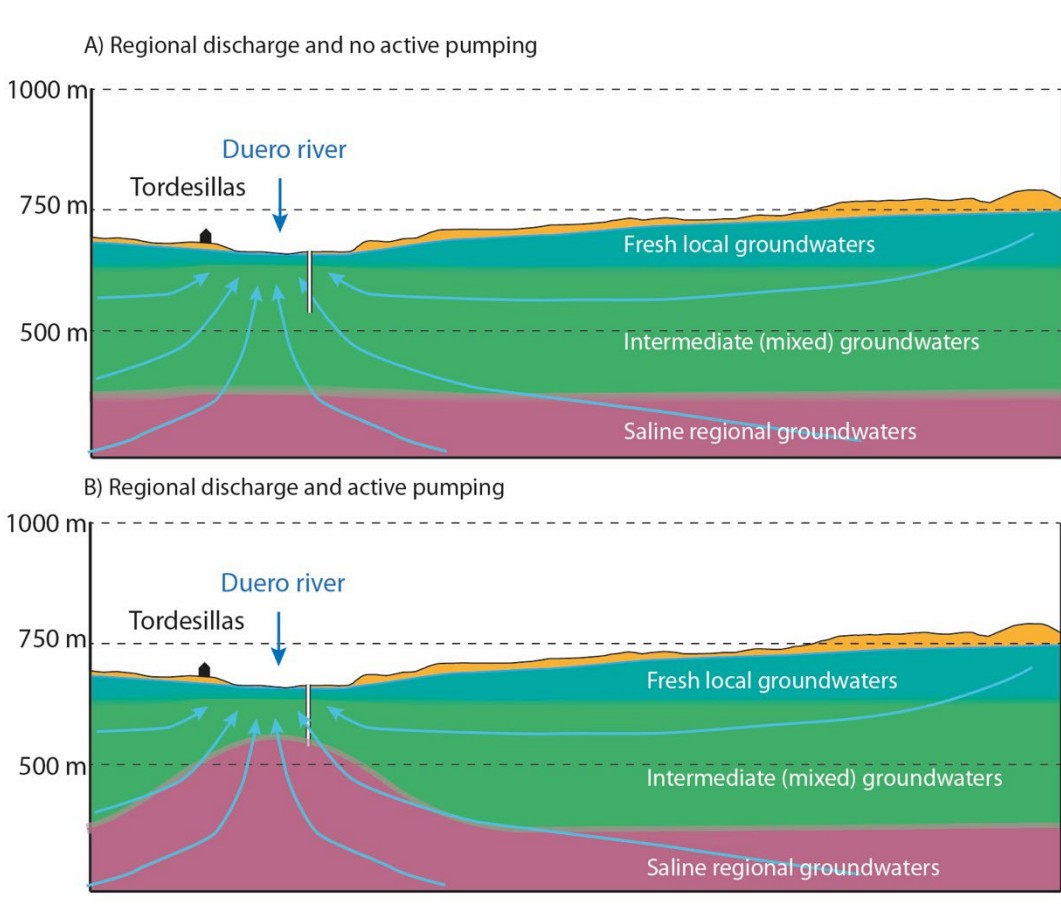


**Figure 9: Conceptual model explaining the increase in salinity during the irrigation campaign. A) Sketch showing the regional discharge in the moments with no active pumping and B) Sketch showing the rise of the saline regional groundwaters at times of active pumping.**

### 6.4 Isotopic evidence of regional groundwater contribution to salinization

Groundwater chemistry in Tordesillas reveals the occurrence of Na- HCO$_3^-$-Cl groundwater most frequent in boreholes drilled within the Duero river floodplain (Fig. 6). In contrast, Ca-HCO$_3^-$ groundwater appears mainly in boreholes located outside the Duero floodplain which intercepts groundwaters with shorter flow paths and which consequently are less saline (Fig. 6; Table C1). Groundwaters from boreholes (both within and outside the floodplain) have lighter $\delta^{18}$O and $\delta$D values than current local precipitation (Valladolid Meteorological Station, 2000-2016) (Fig. 8A; Table C2), indicating that deep groundwaters are not related to local-present-day recharge, but were instead sourced from areas characterized by lower temperatures/higher altitude (specially the lightest values). Comparison with groundwater from boreholes in Villafáfila (50 km to the north) (Huerta et al.,



2021) indicates that both are lighter than local meteoric waters, and correspond to regional groundwaters. Radiocarbon ages from regional groundwaters in Villafáfila indicate 20-30 Ky residence times (Huerta et al., 2021). $\delta^{18}O$ and $\delta D$ values measured

in Tordesillas boreholes are higher than the average of Villafáfila´s regional groundwaters but lighter than current local precipitations, thus suggesting mixing of regional and local groundwaters. Similar interpretations of $\delta^{18}O$ and $\delta D$ isotopic values have been used to discriminate local and young from regional and old groundwaters elsewhere (Guo et al., 2015; Huang and Pang, 2010).

$\delta^{18}O$ and $\delta^{34}S$ of groundwater dissolved sulphate can be used to trace the origin of this anion (Banks and Boyce, 2023; Eastoe

et al., 2022; Jakóbczyk-Karpierz and Ślósarczyk, 2022). Isotopic data from Tordesillas groundwaters (sampled in boreholes, dug wells, and waters from the Duero river) have been compared with published data from Triassic gypsum (Utrilla et al., 1992) and Villafáfila regional groundwaters (boreholes) (Huerta et al., 2021) (Fig. 8B). $\delta^{18}O$ values from dissolved sulphate in Tordesillas boreholes (14.34-17.06 ‰) are similar to those from Villafáfila, but $\delta^{34}S$ data range wider (10.51-17.63 ‰) (Fig. 8B; Table C2). Triassic gypsum has similar to somewhat lower $\delta^{18}O$ (8.9-13.9 ‰) and similar average, but less variable, $\delta^{34}S$

(10.2-16.3 ‰) than Tordesillas groundwaters from boreholes. The above suggests that Tordesillas regional groundwaters have received dissolved sulphate from different sources and measured values reflect mixing from different end-members. The $\delta^{34}S$ values of the dug wells in the Tordesillas floodplain are the lightest of all the measured samples. This could be interpreted as a contribution of sulphate fertilizers, which can provide a wide range of $\delta^{34}S$ values (Spoelstra et al., 2021); the lighter values correspond to anthropogenic sulphate derived from pyrite oxidation in fertilizers (Berner, 1985; Moncaster et al., 2000). The

sulphate isotopic values of waters from the Duero river itself are interpreted as the mean value of all of the sulphate contribution in the Duero basin and fit within the field of the Triassic gypsums (Fig. 8B). A characterization of sulphate isotopes of Cenozoic gypsum formations would help understanding how the different sulphate sources contribute to natural groundwaters.

**7 Conclusions**

Crop salinization by irrigation with groundwaters in the Tordesillas area has been studied to understand the origin of the

problem. The hydrogeological, geophysical and geochemical techniques employed have allowed to obtain the following conclusions.

Regional groundwater flow in the Cenozoic aquifer system of the Duero basin discharges in the Tordesillas area (Duero river). Below depths of 80-160 m (580-500 m.a.s.l.), groundwater salinity (TDS) is about 5000 mg/L as revealed by the TDEM and salinity profiles (Figs. 4 and 5) and progressively decreases upwards by mixing with the discharge of local and less saline

flows. Piezometry reveals and upward component of the groundwater flow (Figs. 2 and 3). Regional groundwater infiltrated at higher altitudes and/or lower temperatures, as evidenced by $\delta^{18}O$ and $\delta D$ isotopic ratios (Fig. 8A) and mixes with groundwaters with shorter flow-paths. This is similar to what has been observed in regional groundwater flows in the Villafáfila area. The wide range of $\delta^{34}S$ and $\delta^{18}O$ in groundwater dissolved sulphate (Fig. 8B) suggests the contribution of different sulphate sources.



TDEM profiles reveal some faults that affect the Cenozoic succession of the Duero basin (Fig. 4A). One of these faults, that coincides with the Zapardiel river, sinks the Miocene succession in the eastern block. As a result, less saline groundwaters in the Paleogene part of the aquifer occur below more saline groundwaters that appear at the base of the Miocene part of the Cenozoic aquifer system.

Crop salinization by irrigation returns is discarded. The $\delta^{18}O$ and $\delta D$ measured on groundwater from the irrigation boreholes doesn't show any evaporation trend. Indeed, the upward groundwater flow component in this area and the higher hydraulic conductivity of the Quaternary aquifer prevents the irrigation returns.

The origin of the crop salinization in Tordesillas area is related to intense groundwater pumping from boreholes used for irrigation. These boreholes, located in the Duero river floodplain, are screened at depths between 100 and 150 m. Pumping is active from spring to the end of the summer. The record of three piezometers screened at shallow (37-70 m), middle (98-190 m) and deep (290 m) parts of the aquifer evidence that during pumping periods hydraulic potential at middle parts decreases drastically favoring the upwelling of more saline groundwaters (Fig. 9). The comparison of salinity profiles along the irrigation campaign confirms the rise of the interphase between less and more saline groundwater.

Salinization in the Tordesillas area is a problem that needs to be addressed by groundwater management authorities to prevent future worsening. The data from this study should help improving the design of pumping wells and determine the depths at which water unsuitable for irrigation is found.





## 8 Appendix A

| Sample name | Date | Sample type | pH | TDS (mg/l) | EC µS/cm 25°C | Ca (mg/L) | Mg (mg/L) | Na (mg/L) | K (mg/L) | HCO3- (mg/L) | SO4 (mg/L) | F (mg/L) | PO4 (mg/L) | NO3 (mg/L) | Fe (mg/L) | Mn (mg/L) |
|---|---|---|---|---|---|---|---|---|---|---|---|---|---|---|---|---|
| P-100 | 09/03/2020 | Boreholes > 40 m | 8 | 483 | 515 | 42.2 | 26.7 | 32.9 | 4.6 | 206.2 | 70.4 | 0.62 | | | 2.9049 | 0.0708 |
| P-101 | 12/03/2020 | Boreholes > 40 m | 7.9 | 720 | 865 | 28.7 | 16.2 | 127.2 | 2.5 | 345.8 | 95.7 | 0.34 | | 29.1 | 0.9318 | 0.0073 |
| P-102 | 12/03/2020 | Boreholes > 40 m | 7.7 | 457 | 672 | 74 | 31.2 | 16.5 | 2.2 | 305.1 | 3.4 | | | 9.5 | | |
| P-12 | 06/04/2022 | Pond | 9.4 | 3232 | 5307 | 10.77 | 4.5 | 1251 | 3.2 | 123.5 | 30.1 | | | 3.5 | | |
| P-28 | 06/04/2022 | Dug wells < 40 m Floodplain | 7.1 | 5002 | 7803 | 461.2 | 107.1 | 1305 | 5.1 | 393.3 | 106 | | | 89.2 | | |
| P-29 | 06/04/2022 | Dug wells < 40 m Floodplain | 7.5 | 2830 | 4314 | 379.9 | 134.3 | 517.3 | 3.4 | 307.9 | 167 | | | 210 | | |
| P-30 | 06/04/2022 | Dug wells < 40 m Floodplain | 8.1 | 340 | 427 | 21.59 | 3.4 | 94.9 | 1.1 | 151.4 | 18.2 | 0.62 | 0.172 | 9.06 | | |
| P-31 | 06/04/2022 | Boreholes > 40 m Floodplain | 8.7 | 308 | 387 | 10.17 | 2.1 | 95.2 | 1 | 127.1 | 14.4 | | 0.152 | 5.86 | 1.024 | |
| P-33 | 06/04/2022 | Boreholes > 40 m Floodplain | 9.5 | 420 | 521 | 3.08 | | 145.6 | 0.9 | 137.2 | 19.5 | | 0.098 | 1.13 | 3.165 | 0.012 |
| P-42 | 06/04/2022 | Boreholes > 40 m Floodplain | 10.3 | 1967 | 2763 | 2.82 | 0.6 | 679.5 | 3.6 | 59.9 | 516 | 1.89 | 0.149 | 1.28 | 0.658 | |
| P-43 | 09/03/2020 | Boreholes > 40 m Floodplain | 9.6 | 669 | 1401 | 3.1 | 1.2 | 229.3 | 2 | 195.4 | 62.6 | 1.63 | | 9.4 | 2.0933 | 0.071 |
| P-43 | 06/04/2022 | Boreholes > 40 m Floodplain | 9.8 | 309 | 441 | 3.9 | | 94.3 | 14.3 | 49.6 | 24.7 | | | 0.25 | 4.865 | 0.024 |
| P-44 | 06/04/2022 | Boreholes > 40 m Floodplain | 8.2 | 1591 | 2274 | 32.24 | 10.9 | 515.4 | 3.8 | 247.9 | 340 | 0.83 | | 2.98 | | |
| P-44 | 06/04/2022 | Boreholes > 40 m Floodplain | 8.1 | 692 | 863 | 38.12 | 9.5 | 180.7 | 2.8 | 325.6 | 53.8 | | 0.171 | 2.49 | | |
| P-55 | 09/01/2020 | Dug wells < 40 m Floodplain | 7.9 | 261 | 294 | 50.1 | 7.1 | 5.8 | 3.7 | 140.8 | 37.5 | 0.15 | | | 4.327 | 0.201 |
| P-59 | 06/04/2022 | Boreholes > 40 m Floodplain | 8.2 | 2862 | 4316 | 41.17 | 23.5 | 991.1 | 6.3 | 167.8 | 749 | 1.48 | | 4.42 | 1.261 | 0.016 |
| P-71 | 06/04/2022 | Boreholes > 40 m Floodplain | 9.3 | 1530 | 2245 | 6.27 | 3.4 | 528.6 | 2.6 | 155.8 | 377 | 2.44 | | 0.5 | 0.374 | |
| P-90 | 15/03/2020 | Boreholes > 40 m | 7.8 | 628 | 681 | 75.4 | 42.9 | 18.9 | 9.6 | 361.4 | 31.4 | 0.41 | | 61.2 | 6.1 | 0.022 |
| P-91 | 15/03/2020 | Boreholes > 40 m | 8.1 | 344 | 428 | 25.8 | 29.6 | 22.4 | 5.1 | 200 | 7.8 | 0.407 | | | 1.91 | 0.015 |
| P-92 | 15/03/2023 | Boreholes > 40 m | 8.3 | 499 | 539 | 14.6 | 10.6 | 104.2 | 1.2 | 309.8 | 12.8 | 0.886 | | 0.7 | 1.31 | 0.0118 |
| P-93 | 11/03/2023 | Boreholes > 40 m | 8 | 606 | 688 | 78.4 | 21.9 | 55.3 | 3.7 | 312 | 33.8 | 0.205 | | 50.2 | 0.1673 | 0.0012 |
| P-94 | 16/03/2023 | Boreholes > 40 m | 7.8 | 672 | 747 | 97 | 25.4 | 37.2 | 2.9 | 387.5 | 32 | | | 43.2 | 0.0308 | 0.1043 |
| P-95 | 16/03/2020 | Boreholes > 40 m | 7.5 | 168 | 253 | 7 | 1.5 | 39.1 | 7.3 | 36.7 | 24 | | | 2.1 | 0.784 | 0.0181 |
| P-96 | 12/03/2020 | Boreholes > 40 m | 7.9 | 580 | 877 | 86.9 | 35 | 50.7 | 2.8 | 305.5 | 28.4 | 0.23 | | 30.8 | 0.0135 | |
| P-97 | 15/03/2020 | Boreholes > 40 m | 7.7 | 519 | 970 | 36.5 | 14.4 | 99 | 17.2 | 136.8 | 3.2 | | | 0.5 | 4.4799 | 0.2575 |
| P-98 | 11/03/2020 | Dug wells < 40 m Floodplain | 9 | 1037 | 1616 | 8 | 25.1 | 293.8 | 6.8 | 119.6 | 267.5 | | | 0.8 | 3.9757 | 0.1365 |
| P-99 | 16/04/2020 | Dug wells < 40 m Floodplain | 8 | 1708 | 2070 | 103.6 | 45.6 | 349.3 | 17.4 | 209.5 | 775.2 | | | 6.9 | 6.02 | 0.0943 |

**Table A1: Groundwater chemical and physicochemical parameters measured in Tordesillas.**



| Sample | Sample type | Water | | Dissolved sulphate | | | |
|---|---|---|---|---|---|---|---|
| | | $\delta^{18}O_{VSMOW}$ | $\delta D_{VSMOW}$ | $\delta^{34}S_{CDT}$ | Yield | $\delta^{18}O_{VSMOW}$ | $\pm 1\sigma$ |
| M-1 Río Duero | Duero river | -7.57 | -54.6 | 12.41 | 76.0 | 12.18 | 0.27 |
| M-2 Río Duero | Duero river | -7.38 | -53.0 | 12.35 | 72.1 | 11.24 | 0.15 |
| P-06 | Boreholes > 40 m Floodplain | -8.23 | -60.5 | | | | |
| P-12 | Pond | -1.12 | -9.7 | | | | |
| P-13 | Boreholes > 40 m Floodplain | -8.47 | -63.1 | | | | |
| P-28 | Dug wells < 40 m Floodplain | -8.15 | -60.0 | 8.77 | 50.8 | 12.12 | 0.26 |
| P-30 | Dug wells < 40 m Floodplain | -8.66 | -65.1 | 10.36 | 65.2 | 10.25 | 0.34 |
| P-31 | Boreholes > 40 m Floodplain | -9.24 | -69.2 | | | | |
| P-42 | Boreholes > 40 m Floodplain | -8.97 | -65.6 | 17.63 | 75.6 | 14.34 | 0.08 |
| P-43 | Boreholes > 40 m Floodplain | -8.55 | -63.2 | 10.51 | 55.6 | 17.06 | 0.02 |
| P-44 | Boreholes > 40 m Floodplain | -8.66 | -64.4 | 13.38 | 31.7 | 15.69 | 0.02 |
| P-58 | Boreholes > 40 m Floodplain | -8.23 | -61.9 | | | | |
| P-59 | Boreholes > 40 m Floodplain | -9.74 | -71.7 | 13.58 | 95.8 | 14.39 | 0.14 |
| P-77 | Boreholes > 40 m | -8.09 | -60.7 | 17.18 | 60.6 | 16.99 | 0.28 |
| P-97 | Boreholes > 40 m | -9.35 | -68.1 | | | | |

**Table A2: Isotopic data measured in Tordesillas groundwaters and the Duero river.**

**9 Author contributions**

ER, LL, PH and PC participated in fieldwork, ER, PH and LL conceptualized the objectives and methodology of this research and prepared the figures and tables. PH found the funding for the project and prepared the original draft. CR participated on the isotope analysis, discussion and draft revision.

**10 Competing interests**

The contact author has declared that none of the authors has any competing interests.

**11 Acknowledgements**

Authors acknowledge A. Blanco Coronas for her help during fieldwork and data processing.



**12 Financial support**

This paper is part of the research project: PID2022-140713NB-I00 and PGC2018-094566-B-C21 funded by the Ministerio de
Ciencia, Innovación y Universidades, MCIN/AEI/10.13039/501100011033/FEDER, UE.

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
