# Peer review of "Crop salinization by intense pumping in regional discharge areas of an inland aquifer system (Cenozoic Duero basin aquifer, Spain)"

_EGUsphere, 2025_

## Author Comment (AC1)

Referee comments RC-1

*Unfortunately, this paper is not suitable for publication in an international journal such as HESS. It has no novel aspects and would not be of interest to the broader academic community. The conclusions are reasonable, but the main finding that groundwater pumping causes changes to aquifer salinities is not unexpected and not novel. In its present form it is more suited to a regional journal. Case studies are certainly acceptable in the international literature, but they need to add to our general understanding; so, indications of what new ideas come out of a study and what researchers working on similar projects elsewhere can take from it are needed.*

**RESPONSE**

We cannot agree with the comment above by RC-1, although admittedly we should have presented our aims, as well as the hydrogeology of this part of the Duero basin, in a better way, which we will do in the revised version of the manuscript.

Certainly, the case is not that groundwater pumping causes changes to aquifer salinities, which, as indicated, is not unexpected. The real issue, in such cases, is to determine the provenance of saline waters, and ascertain the reasons (anthropic? geological? structural? …?) why those crop out now but didn't do so in the past. And those findings certainly have the potential to be of broad interest not only to the academic community, but for the wider society as well. To that end, the manuscript characterizes how regional groundwaters discharge in one of the largest aquifer systems in the Iberian Peninsula using geophysical tools on the Duero river discharge zone that show the distribution of salinities in the area. Hydrogen and Oxygen stable isotopes allow discerning between different regional groundwaters and sulphate-S and -O isotopes help to understand where the salts were acquired. How the combined use of different techniques helped to understand the hydrogeological behaviour of a large aquifer system it is not only important to Iberian researchers, but it is also a good example for worldwide researchers on how the saline groundwater flow, in a large aquifer system disconnected from the marine realm, discharges through a river, given insight into the origin of solutes, and illustrating how the geology and structure of the basin determine groundwater quality and flow.

That saline water eventually affected crops is anecdotical, and was mentioned as an indication of how a local problem triggered a scientific enquiry with much broader implications.

*The limited and parochial scope of the paper is evident in the Introduction, which is focused on the local issue of crop yields being impacted by salinization. Salinization of water resources is a major problem globally, but there is no effort here to review the global understanding or to put this study into a broader context. The Aims are also very specific to understanding the local hydrogeology and the Conclusions are just a restatement of the specific findings of the research with no indication of how or why this research is of general interest. Even within the local context, the end of the paper is underwhelming with a general suggestion that these data should help management (without specifying how).*

**RESPONSE**

We agree with the Reviewer that our choice of wordings for the introduction section was not the most fortunate, and we did not manage to properly get our message through as a result. In the revised version of the manuscript we have modified the introduction to give a broader perspective on the problem. Papers like that of Thorslund et al. (2021, cited in our manuscript), that review the role of irrigation drivers in salinization, require of regional examples to give a global perspective. Our manuscript is not focused on understanding the local hydrogeology; rather it characterizes the discharge area of one of the largest aquifer systems in the Iberian Peninsula and how different groundwater masses respond to varying demands. We will modify the aims of the paper pointing the importance of salinization in discharge areas, and the broad perspective gained by combining different hydrogeological techniques. This can be useful to other researchers to see how the combination of different approaches help to understand hydrogeological earth systems and their processes. For example, the Reviewer mentions below the case of the Murray Basin of Australia (as described in Herczeg et al., 2001). The situation on the continental Duero Basin dealt with by our research might be a suitable example of how a similar problem (groundwater salinity) may have contrasting causes in very different geographical and geological contexts, and at such different scales (more on that below).

*There are several issues with the data and its interpretation.*

1. ***Data limitations.*** *The geochemical interpretations are based on a standard set of parameters (major ions, water stable isotopes, sulfate stable isotopes) from a small number of groundwater samples (12 in total). This is a very limited dataset that is unlikely to yield much insight into processes. The groundwater samples are also poorly characterized – the text and Table A1 lists samples as being <40 m and >40 m, but what are the exact depths? It is much more straightforward to interpret geochemistry data from wells that have short screens and which sample water from a specific aquifer*

*than from wells with long screens that integrate water from several layers. Section 5.2.1 divides the sequence up into several GU's and it would be good to be able to link the geochemical data with those units, which is not possible with the current reporting.*

**RESPONSE**

Every researcher wants to generate and deal with as much data as possible, and we are not any different in that regard. However, all too frequently we have to adapt to reality, and try to make the best out of limited resources (because of funding, suitable samples and sampling points, access to analytical facilities, …). Perhaps the number of analyses could be larger, but we report 27 geochemical analyses of major and minor elements, 15 $\delta D$ and $\delta^{18}O$ measurements in water , and $\delta^{34}S$ and $\delta^{18}O$ in sulphate. For the period studied the number of samples, is limited by the number of observation wells (piezometers from the Water Authority) and other private pumping wells located in the area. Wells outside the regional groundwater discharge area have not been taken into account, although most of them show similar groundwater chemistry than reported boreholes located outside the floodplain (boreholes > 40 m, in the manuscript). We have also cited work from other authors working in the area, as well as from our previous publications. This makes the data set more robust. We classify the different wells in three categories: 1) Dug wells; 2) Boreholes > 40 m; 3) Boreholes > 40 m located within the floodplain. We make this classification by two reasons: A) because it helps to simplify the explanations and to understand better what happens in the shallow, mid-upper, and deep parts of the aquifer. B) Because there is no information about the screened depth of the private irrigation wells sampled. Dug wells normally are excavated to no more than 4 m in the floodplain because they inundate fast as they are constructed. None of them reaches depths of tens of m. In contrast, irrigation boreholes are commonly screened between 100 and 150 m depth. The arbitrary choice (shallower / deeper than 40 m) of depth was made to clearly differentiate between shallower and deeper waters. In the revised manuscript we will add the screened depth of some piezometers (when known) and an interrogation sign for those without depth data.

Regarding the GUs mentioned in section 5.2.1 it should be noted that these are defined as a result of our investigation; these were not know "a priori", so no targeted sampling was possible at the time. To improve Section 5.2.1 in the revised manuscript, we will draw P42 (290 m), P43 (34-70 m) and P44 (98-190 m) on Fig. 4 profile1. These piezometers locate in the same place than SEDT-4. This will link the different Geoelectrical Units to at least some of the piezometers sampled.

2. ***Major ions.*** *The interpretation of the major ion geochemistry does not really add much. Section 5.3 is descriptive and mainly defines groundwater types and their distribution. Section 6 mainly makes use of the overall groundwater TDS and salinity but not really the major ion geochemistry. Mixing is discussed in Section 6.2, but there is little attempt to quantify it or justify the conclusions. Even if quantification is not possible, identifying the end-members and showing the mixing on the Piper diagram or other plots would help. However, it is difficult to use small datasets to produce robust conclusions about processes such as mixing, which makes this discussion speculative. Some of the changes in water chemistry may reflect processes such as mineral dissolution and precipitation, which can produce systematic changes in water chemistry with salinity (e.g., Herczeg et al., 2001. Origin of dissolved salts in a large, semi-arid groundwater system: Murray Basin, Australia. Marine and Freshwater Resources, 52, 41-52, https://doi.org/10.1071/MF00040); again, this needs consideration.*

**RESPONSE**

We intended to make section 5.3. descriptive; it is important to separate descriptions from interpretations. Major ion geochemistry allows us to separate the facies types in the Duero discharge area, and these will be used in the discussion (Section 6).

We agree with the Reviewer that more discussion is needed, combining major ion geochemistry, TDS and isotopic results. We will include in section 6 comments regarding these. Mixing of two different end-members has been identified in the area: the Na-Cl regional groundwaters sampled mainly from boreholes located in the floodplain. This is consistent with the results obtained from other areas of the Duero aquifer system, where deep saline groundwaters occur (Huerta et al., 2021). The other endmember are the local groundwaters with Ca-HCO$_3^-$ compositions. Groundwaters of this type are identified in the hills or in shallower parts of the aquifer system in other areas. The data set reported is limited to the study area and to the period we pretended to explore. But previous published results and publicly available data, accesible in in different open repositories (https://info.igme.es/bdaguas; https://mirame.chduero.es/chduero/public/home) confirm the occurrence of these two end members (local and regional groundwater flows).

We will modify the Piper diagram to show the mixing process.

The combination of the major ion geochemistry, piezometry, isotopic composition and geophysical results supports that our interpretations are robust. Check the

progressive salinity decrease (geophysical profiles and EC variations in piezometers) in a discharge area with an upwards component of the groundwater flow.

Changes in water chemistry in the area studied are not related to mineral dissolution and precipitation processes. Rather they are simply related to different residence times of the groundwaters (local Ca-HCO$_3^-$ vs regional Na-Cl types). Obviously the different compositions are controlled by dissolution and precipitation processes that occur along the flow paths as proposed by Chebotarev, 1955 and many other researchers along the world. See cites in the introduction of Heczeg et al, 2001. On the other hand, there are many differences with the Murray basin studied by Heczeg et al, 2001. The scale, the height and location respecting the coast, and the geological history of the basin. In the case of the continental Duero basin, there are no marine evaporites within the basin fill, nor any possible marine groundwater contribution. Outcrops in the mountain ranges surrounding the basin are dominated by igneous and metamorphic rocks and clastic and carbonatic sedimentary rocks. Marine Mesozoic evaporites are minor, although these contributed solutes for the Cenozoic non-marine deposition of sulphates (mainly gypsum) (Huerta et al., 2010) doi:10.1016/j.palaeo.2009.12.008 .

3. ***Water stable isotopes.*** *The comparison of the groundwater stable isotopes with those of the weighted mean precipitation (Section 5.4) implicitly assumes that the groundwater is recharged by precipitation with that isotopic composition and any deviations occurs due to recharge at elevation (Section 6.4). However, it is likely that recharge occurs preferentially from high rainfall events or during winter when evapotranspiration rates are low and the comparison needs to consider this. Is it possible that the isotopic composition of the rainfall that causes recharge is different from the annual mean – for example, recharge mainly from winter rainfall, which commonly has lower stable isotope values than the mean, may explain the observations.*

**RESPONSE**

We are sorry to notice that the Reviewer seems to have misunderstood our point here. In fact, our argument is that isotope ratios measured in deep groundwaters are at odds with the possibility of an origin by local recharge.

It is common wisdom that "most groundwater bodies are isotopically constant and closely reflect the average annual isotopic composition of local precipitation" (Fritz, P. (1971), IAEA Tech. Rep. Ser. #210; pg. 179; see also Gat, J.R. in the same volume, as well as compilations in classical reference books such as Clark & Fritz, 1997,

"Environmental isotopes in hydrogeology", Lewis Pub., or Drever, J.I., 1997, "The geochemistry of natural waters. Surface and groundwater environments". Prentice Hall). In small river catchments seasonal variation of stable isotopes in precipitation can be reflected by river water, but these are smoothed out in larger rivers (such as Duero), that tend to approach the isotopic values of the weighted average precipitation at their catchment (see, for example, Kendall, C., and T. B. Coplen (2001), "Distribution of oxygen-18 and deuterium in river waters across the United States". Hydrol. Processes, 15(7), 1363–1393, doi:10.1002/hyp.217.); Bowen, G.J. et al., 2011, "Water balance model for mean annual hydrogen and oxygen isotope distributions in surface waters of the contiguous United States". J. Geophys. Res., 116, G04011, doi:10.1029/2010JG001581, 2011). Unpublished data for the Tormes river, a major tributary to the Duero river from the south, sampled on a weekly basis along several uninterrupted months, and the weighted average of monthly values of precipitation in a reference meteorological station located towards the centre of its catchment for the same period do coincide, thus reassuring us that our reported values for the Duero river waters, as well as those of the met stations used as reference do make sense.

Certainly, we do not have data detailed enough to characterize isotopic values of long-term precipitation over the whole Duero catchment, but the fact that average precipitation at Valladolid Met Station for the period 2000-16 is only marginally lower than the river water, and that both are obviously higher than values measured in deep groundwaters is, in our opinion, a strong indication that deep groundwaters are not simply infiltrated local meteoric waters, and this even before taking the high salinities of deep groundwaters into consideration

4. ***Water stable isotopes.*** *The observation that the stable isotopes do not define an evaporation trend may not rule out irrigation returns. Open-system evaporation in surface water bodies (pools, lakes etc) does produce distinctive isotopic trends. However, transpiration does not and it is not clear whether evaporation from within the soils where the relative humidity is higher fractionates stable isotopes to a large degree. There are plenty of examples of saline groundwater caused by evapotranspiration where the stable isotopes lie close to the MWL. This would include the deep groundwater in this region – looking at the major ion geochemistry and description of the aquifers, the high salinities are probably the result of evapotranspiration (there are no evaporites reported and halite dissolution produces a distinct NaCl-type geochemistry that is different to what is shown in Fig. 6). Yet the stable isotopes lie very close to the MWL. Again, the interpretation of data needs to be better justified.*

**RESPONSE**

Shallower samples (those from dug wells P-28 and -30) plot alongside deeper samples from boreholes, and all of them extrapolate, on the heavy side, towards the values measured in Duero river, which are not that far away from average local meteoric precipitation. We are aware of the position defended by the Reviewer that groundwater may represent (mostly?) infiltration of winter precipitation, thus isotopically lighter that the weighted average, although that would not explain the values of the river water. But even accepting the above, some of the deep, saline waters are very light, and extrapolating the observed correlation towards the light side leads to values very close to those reported for fluent deepwaters at Villafáfila (see Huerta et al., 2021, cited in the references), also saline. There is no irrigated agriculture in the wide vicinity of Villafáfila, so irrigation returns there are unlikely, and, moreover, there is both archaeological (Bronze age) and historical evidence of exploitation of such waters as a source of salt. Radiocarbon dating of some of these saline groundwaters resulted in ages in excess of 20.000 yBP. S of the Duero river, in Medina del Campo, some 25 km S of our study area, there is a historical (XIX century) Spa for which a salinity of 72 g/L was reported in analyses of 1892, well before groundwater was used significantly for irrigation (pumping started in the '70s of the XXth century). Similar waters are documented in Olmedo, some 35 km SE of our study area. Please see the article by de la Hera-Portillo et al. (2021), cited in the reference's list.

So, we think that we have sound reasons to believe that high salinity is a characteristic intrinsic of deep groundwaters, and not a recent product of irrigation returns.

5. **Sulfur isotopes.** *The discussion of the sulfur stable isotopes (section 6.4) is also general and not well justified. The isotopic values are interpreted as solely representing gypsum dissolution in the aquifers without consideration of whether other sources of sulfur (e.g., pyrite) might be present or whether fractionation due to processes such as bacterial sulfate reduction (which is common in groundwater globally) may have occurred. The conclusion that the Tordesillas groundwaters have sulfate derived from several sources is untested (is that consistent with the other data and the hydrology?). Similar comments apply to the conclusion that the isotopes show mixing in the river. As with the other datasets, you need to justify potential interesting conclusions such as this rather than just making assertions.*

**RESPONSE**

We have difficulties following the reasoning of the Reviewer regarding sulphur isotopes when considered at the light of his/her earlier comments: the Reviewer has argued before against a "distal" provenance of groundwater, favouring infiltration of the lightest component of local precipitation instead. The Tertiary, continental, sediments infilling this part of the Basin are essentially devoid of sulphides. Certainly, there are abundant sulphides (pyrite, but also monosulphides -i.e., pyrrhotite-) in the Variscan metasediments of the western margin, but there is no hydrological evidence of substantial contribution of waters coming from the western margin to this part of the Duero Basin. Even if such was the case, $\delta^{34}S$ measured on several tens of sulphide samples roughly group around two contrasting values; <+5‰ in older (Neoproterozoic?) metasediments and <-10‰ in materials closer to the Precambrian/Cambrian boundary age.

In contrast, sulphates, such as those included in Fig. 8 for reference, are not unusual in the Basin margins, including those areas contemplated as possible recharge areas for the deep groundwaters in our hypothesis. Moreover, occasional (mostly diverse forms of gypsum) sedimentary Tertiary sulphates within the Basin's infilling have $\delta^{34}S$ values between +14.3 - +15.7‰ (Armenteros and Recio, 1995; XIII Congreso Español de Sedimentología, Teruel), also compatible with the data in Fig. 8. More extensive surveys, currently under way, of Tertiary sulphates sampled from the infill of the Duero Basin so far give $\delta^{34}S$ = 15.0±1.6‰, n = 82; $\delta^{18}O$ = 17.8±3.6‰, n=45; not that different from those reported for dissolved sulphate in the deep groundwaters studied. Please note that no extremely heavy nor extremely light isotopic values have ever been found.

Should sulphate reduction had contributed (significantly?) to the dissolved sulphate in deep groundwaters, (very) high $\delta^{34}S$ would be expected in residual sulphate, as well as dissolved $HCO_3^-$ / carbonate cements characterized by very light $\delta^{13}C$. None of such has been identified, and the few analytical determinations of dissolved oxygen available indicate that waters are oxidant. Using the Murray Basin as reference, derived from the Reviewer's mention of the article by Herczeg et al. (2001), Dogramaci et al. (App. Geochem., 16, 475-488, 2001) report abundant $\delta^{34}S$ and a few $\delta^{18}O$ (19 data; not that many more than ourselves) values for groundwater from the Murray Basin. From those data, Dogramaci et al.'01 hypothesize five possible sources of sulphate to groundwater; i.e. atmospheric deposition, marine aerosols, dissolution of sulphate minerals, oxidation of sulphides and mineralisation of organic S.

Considering the values they report, and the geological and geographic context of the MB, these potential sources make sense and are worth discussing, but that is not extrapolatable to the Duero Basin due to its geological nature and history, and its geographical location:

Atmospheric deposition of sulphate is unlikely. Climatological features show that dominant winds in the region are westerlies to north westerlies, and there are no evaporites within the wind trajectory that could contribute dust, nor significant urban / industrial areas that could contribute $SO_2$ pollution (data available indicate <25 ppb average annual $SO_2$ concentration; see Schleicher and Recio, 2010, Environ. Sci. Pollut. Res., 17, 770-778).

Sea spray was also discarded by Schleicher and Recio (2010) as a significant contributor at Salamanca, some 300 km away from the nearest coast in the dominant wind direction. The area of study, in the vicinity of Tordesillas, is some ~75 km NE of Salamanca, so even further away from sea spray sources.

The sedimentary infill of the Duero Basin is essentially devoid of sulphides, so oxidation of sulphides is an unlikely source of sulphate to the groundwaters. The nearest rocks with significant sulphides are in the western margin of the basin, and oxidation of the sulphides present give lower values (Jambrina et al., J. Limnol, 72(2), 361-365, 2013) than measured in dissolved sulphate. In any case, on hydrogeological grounds, such area does not contribute to the groundwaters under consideration.

As for oxidation of sulphides, there is not significant organic matter within the sediments of the Duero Basin in the area, nor in the potential source region of analysed groundwaters, so mineralisation of organic S is an unlikely contributor to the sulphate pool as well.

As a result, dissolution of preexisting sulphate minerals is the only reasonable potential source of dissolved sulphate to the "deep" groundwaters analysed ("shallow" groundwaters may have additional contributions, as discussed in the manuscript).

It would seem that the Reviewer misunderstood our comment regarding the mixed-origin character of sulphate analysed from the river. The Duero river collects water contributed by its tributaries, and together drain the Basin towards the West. In the area of study, the Duero also represents the discharge area of regional groundwaters. Considering fluvial dynamics, it seems reasonable to consider it to be well mixed, and given that isotopic values are intermediate between those of saline deep waters, and those of shallower

groundwaters from agricultural areas, to conclude that measured values may be product of mixing from different sources seemed natural. We do not argue that isotopic values demonstrate mixing; rather we acknowledge that due to fluvial and groundwater dynamics contribution from multiple sources, should those be available, is highly likely.

6. ***Integration with the geophysics data.*** *Partially due to the lack of detail regarding sample depth, it is difficult to link the geochemistry with the geophysics data. The geophysics results (discussed in Section 6.1) are presented separately to the geochemistry. Integrating both halves of the work would help the study.*

**RESPONSE**

Yes as we have mentioned before we will integrate better the geochemical and geophysical data as the available information allows us.